# Simulation and Test of a Contactless Voltage Measurement Method for Overhead Lines Based on Reconstruction of Integral Node Parameters

**DOI:** 10.3390/s20010246

**Published:** 2019-12-31

**Authors:** Jingang Wang, Xiaojun Yan, Lu Zhong, Xiaobao Zhu

**Affiliations:** 1State Key Laboratory of Power Transmission Equipment and System Security and New Technology, Chongqing University, Chongqing 400044, China; jingang@cqu.edu.cn; 2Southwest China Branch, State Grid Corporation of China, Chengdu 610041, China; jessie_zhonglu@126.com; 3School of Software, Nanchang Hangkong University, Nanchang 330063, China; zhuza@mail.uc.edu

**Keywords:** contactless measurement, E-field integral method, field-source solution, Gauss–Chebyshev algorithm, parameters reconstruction

## Abstract

To improve the stability and adaptability of the voltage measurement based on the E-field (electric field) integral method, in this paper we introduce a new method for the contactless voltage measurement of the overhead lines. The method adopts the node parameter reconstruction technology, which is based on the Gauss–Chebyshev algorithm. In order to achieve high-quality E-field detection at the reconstructed node position, we designed a novel D-dot sensor with parallel distributed electrodes. A Maxwell simulation model of multi-level voltages of the overhead lines was carried out to determine a comprehensive criterion of the reconstruction factors. The simulation employed a three-phase overhead line experiment platform to calculate and measure the distribution and the changing trend of the E-field. The deviations of the voltage measurement were reduced at a significantly low level within 0.4%. The result of the simulation demonstrates that the method optimizes sensor distribution by reconstructing node parameters, which enables the system to have high accuracy and reliability on the contactless voltage measurement of the overhead lines.

## 1. Introduction

With the rapid construction of the smart grids, the shortcomings of traditional voltage measuring methods in installation, maintenance, and measuring accuracy are increasingly obvious, and can hardly meet the fast, accurate, and stable measuring requirements [1,2,3]. There is no energy transfer between the transmission line and the sensors, which is featured by NDT (nondestructive testing). Hence, the contactless measurement method is extensively studied [4,5,6]. The basic approach of this method is to accurately restore the voltage parameters of the overhead line by the spatial E-field information captured by the contactless sensor. Therefore, the type and design of the sensor terminal, the algorithm, and model of the field-source solution will directly determine the accuracy of the entire measurement system [7].

Among the sensor types, many important achievements have emerged in the research area of optical E-field sensors. A recently published article by Yong Cui et al. [8] introduced a sensor based on the Pockels effect that may detect the distorted E-field better than the Narda-NBM-550 sensor. J. Firth et al. [9] developed a new optical liquid crystal E-field sensor, which can improve the measurement accuracy to 1% even though the strong E-field reaches 380 kV/m. Optical crystal sensors are used in a variety of materials and structures, and have good output response characteristics [10,11,12,13]. However, due to the high precision of the terminal, some performance characteristics are susceptible to the complex application scenarios.

A D-dot E-field sensor features good measuring accuracy, high insulation performance, small volume, and low integration difficulty, which is more suitable for the large-area distribution for the contactless voltage measurement [14,15,16]. Some scholars focus on the design method and the structure optimization of D-dot E-field sensors. Hu Xueqi et al. [17] designed a vertically placed D-dot sensor and proposed a decoupling method for the superimposed E-field. Fan, YY [18] presented an efficient D-dot sensor design method based on a field–circuit coupling design method. Tao Yaqin, Zhao Pengcheng [19,20] utilized a double differential D-dot sensor for high frequency E-field signal measurement. Based on D-dot and B-dot principle, T. Huiskamp [21] combined the function of pulse signal detection for the high-frequency voltage and current. Different sensor designs correspond to the various installation standards. However, the previous studies of D-dot sensors have not dealt with the problem of interference and insulation in voltage measurements. To improve the reliability of E-field detection signals, in our work, a novel D-dot sensor with parallel distributed electrodes is applied to the contactless voltage measurement system.

In the field-source solution methods, the inverse-problem theory is complicated and unstable in voltage calculations. In addition, the accuracy is easily affected by the distribution of measuring points and the environmental factors [22,23,24], so the new numerical analysis is needed to improve the efficiency of the field-source solution. The E-field integral algorithm is an advanced access to solve the problem bases on detecting the E-field information by the multi-node sensors and weighted sum [25]. Such a method has been applied in the optical voltage sensors. Rahmatian F [26] composed a transducer with the three-node optical E-field sensor array and the measurement accuracy can reach the IEC (International Electrical Commission) 0.2 standard after the integral calculation. However, the OVT (Optical Voltage Transducer) is bulky and there is little discussion about the insulation problem about the sensor array. Limited by the complicated structure and high production cost, the optical E-field integral method is difficult to widely promote in the power system [27].

In contrast, D-dot sensors are better suited for miniaturization and integration applications. The accuracy of the D-dot voltage measurement system is also affected by the quantity and distribution of sensors. Zhao Yanhang [28] first attempted to restore the voltage information using D-dot sensors and a Gaussian integral algorithm, and found the relative error of voltage measurement was less than 0.27%. However, the results were obtained under the ideal conditions, and some issues were actually minimized, such as the unstable measurement performance caused by the relatively fixed nodes information and the nonlinear E-field distribution. Li Xiang [29] has improved the Gaussian integration algorithm, in order to obtain better insulation performance of the sensors while ensuring the good measurement accuracy. A flaw is that this research ignored the other interference sources that may be caused when the sensor position moves downward. At the same time, the node selection could not match the parameters of overhead lines, and this will result in the waste of resources and unstable measurement, so the measurement stability and the sensor distribution scheme can be further improved.

In view of this, the main contribution of our work is removing the limitation that the integral node distribution scheme cannot match the overhead line condition, and improving the onsite adaptability of the E-field integral method through the node parameters reconstruction. First, in the second chapter, the new design of the D-dot sensor and the structure of the contactless voltage measurement system are briefly introduced, and the Gauss–Chebyshev integral node parameters reconstruction algorithm is further presented. Then, in the third chapter, the reconstruction factor *k* of the integral domain under the multi-level voltage of overhead line is determined, and the measurement accuracy after the algorithm reconstruction is tested by simulation. In the last section, the measurement effect before and after the node parameters reconstruction is compared to the test result under the three-phase overhead line platform.

## 2. Measurement System and Algorithm Model

### 2.1. Detection Terminal

The accuracy and standardization of E-field detection of sensors on the integral node position directly affect the accuracy of voltage measurement from the source of parameters. Therefore, the principle of field–circuit coupling method is used to design a new version of D-dot E-field sensor as shown in Figure 1. The electrode in the sensor consists of bi-level layers of metal with parallel spiral structure and is equipotential distribution, which makes the sensor have a high sensitivity to fully sense the spatial E-field information, and a good insulating property under the action of the extremely strong E-field.

The width, thickness, and spacing of the upper or lower electrodes of the sensor are 1.0 mm, 0.1 mm, and 1.6 mm respectively. Within 40 Hz^−1^ MHz bandwidth, the mutual capacitance of the terminal parallel electrode measured by the impedance analyzer is 72 pF, and the average resistance value is about 1 MΩ. The main design of microelectrode and sensor reduces the error of integral computation caused by E-field distortion.

The arrangement of D-dot sensors is parallel to the ground potential reference plane which can realize contactless measurement with overhead lines. The proportional relation between the output signal of D-dot sensors and the intensity value of the spatial E-field can be obtained through calibration by the standard electromagnetic field measuring instruments.

### 2.2. Construction of the Voltage Measurement System

The voltage measurement system based on numerical integration is mainly composed of the sensor array, the signal conditioning and digitization circuit, and the integral algorithm routine. The overall structure of the framework is shown in Figure 2.

The sensors are installed in an array with an insulating rod according to the node position information calculated by the integral formula, and the output data packet is transmitted to the PC terminal by the serial port and the WIFI module through differential amplification, low-pass filtering, level lifting, A/D conversion, and other processes.

The upper computer program of the integrating voltage measurement system was designed by LabVIEW software, and the discrete E-field data packets received by the PC were digitally restored through UDP (User Datagram Protocol) protocol. Finally, the E-field integration was carried out to realize the voltage measurement, and the voltage waveform could be displayed on the main program interface UI for the waveform quality analysis.

### 2.3. Node Parameters Reconstruction Algorithm Model

Influenced by the environmental factors, the E-field distribution functions under the overhead lines are quite complicated and only finite sensors can be used for E-field measurement in the actual application. Therefore, it is quite difficult to derive the voltage using the fundamental calculus theorems through acquiring the numerical values of continuous E-field strength.

#### 2.3.1. Gauss–Chebyshev Algorithm

Above the Gauss integral algorithm, the Gauss–Chebyshev algorithm model is more suitable for the calculation [30,31,32]. The principle of Gauss–Chebyshev algorithm is shown in Figure 3. *P*_1_, *P*_2_, and *P*_3_ are the different spatial E-field integral paths between the overhead lines and ground potential surface. If the tangent vector modulus of the partial position field on the integral path can be acquired and the numerical value integration can be conducted along the path direction, the voltage of overhead lines to be measured can be obtained.

The weighted summation of the partial discrete E-field values is carried out to replace the calculation results of the definite integral voltage in the region, and the solution process can be simplified within the error range.

A certain plumb line *l* from the overhead line to the earth can be used as the integral path, and then the wire voltage can be obtained from the Gauss–Chebyshev basic formula:(1)$$\varphi_{H}=\int_{0}^{H}E\left(x\right)dx\approx \overset{n}{\underset{j=0}{\sum }}A_{j}E\left(x_{j}\right)$$
where *φ_H_* refers to the voltage of the wire to be measured. *H* refers to the plumb height between the wire and the ground. *E*(*x*) refers to the irregular E-field function related to the distance to ground, *x*, in the direction of *z*. *x_j_* refers to the position of *n* integral nodes in the interval of [0, *H*]. *A_j_* refers to the corresponding weighted coefficient.

Set *f*(*x*) as the irregular E-field distribution function. By normalizing the upper and the lower limits [0, *H*] of the E-field, the wire height parameter can be converted into the interval [−1, 1]:(2)$$\int_{0}^{H}f\left(x\right)dx=0.5H\int_{-1}^{1}f\left(0.5dt^{\prime }+0.5H\right)dt^{\prime }$$

If the intermediate variable *t* = 0.5*H*(*t*’ + 1) and the number of Chebyshev discrete integral points are selected as *n*, there will be following result in the normalization interval:(3)$$\int_{-1}^{1}\frac{f\left(t\right)}{\sqrt{1-t^{2}}}dt\approx \frac{\pi }{n+1}\overset{n}{\underset{j=0}{\sum }}f\left(\cos\frac{2j+1}{2\left(n+1\right)}\pi \right)$$
where $\frac{\pi }{n+1}$ refers to the integral weight, and the position of the integral node corresponds to the right form equals to $\cos\frac{2j+1}{2\left(n+1\right)}\pi$ Set *t_j_* as the theoretical node and *x_j_* as the normalized node, and the following integral node information in Table 1 can be calculated by the aforementioned algorithm method:

It can be seen that the weights *An* and the position of the *t_j_*/*x_j_* integral node changed with the quantity *n*. The 35 kV overhead lines setting with 7 m minimum safe distance, e.g., all the distribution schemes encounter the problem that the position of some nodes (marked in red), is too close to the ground so the E-field signals detected at the node positions are single, relatively weak, and vulnerable to the other disturbance sources, meaning the system measuring accuracy cannot meet the practical requirements. Hence, limited by the practical measuring conditions, there remains improvement space in the Gauss–Chebyshev algorithm.

#### 2.3.2. Algorithm Model of Node Parameters Reconstruction

In order to solve the aforementioned shortcomings of the Gauss–Chebyshev integral algorithm, the node parameters reconstruction method is proposed, whose principal task is to segmentally partition the integral interval of overhead wires into linear region and nonlinear regions. While the quantity of sensors is limited by the practical application, in order to make the measuring data contain the most E-field distribution information under the lead to be measured, the integral intervals are divided for discussion according to the weights of integral nodes in the numerical sum. This is so the distribution of sensors after algorithm improvement can meet the practical application requirements better, and the measuring accuracy and applicability of system can be improved simultaneously.

Through the reliable reconstruction factors, *k*, the E-field integral path under the overhead wires is divided into the near-ground area and near-source area, which are marked as *L*_1_ and *L*_2_. If *a* = 0 and *b* = *kd*, the normalization of E-field intensity, *f*(*x*), is accomplished to obtain the relationship between *x*_1*j*_ and *d* in the interval of *L*_1_: *x*_1*j*_ = *k*_1*j*_
*d*. The voltage formula obtained through the E-field integration in the interval of *L*_1_.
(4)$$V_{1}=\int_{0}^{kd}f\left(x\right)dx=\overset{m}{\underset{j=1}{\sum }}A_{1j}f\left(x_{1j}\right)$$
(5)$$x_{1j}=0.5kdt_{1j}+0.5kd,j=0,1,2\ldots \ldots m$$
(6)$$A_{1j}=\frac{kd}{m},j=1,2,3\ldots \ldots ,m$$
where *V*_1_ is the numerical integration result of the interval *L*_1_; *m*, *A*_1*j*_, and *x*_1*j*_ refer to the amount, weights and positions of the integral nodes in the interval *L*_1_ respectively; *k* is the reconstruction factor, and *k*_1*j*_ = 0.5*k*(*t*_1*j*_ + 1) refers to the reconstruction coefficient in the *L*_1_.

As for the integral interval of near-source section, *L*_2_, if the quantity of measuring points is *n*, *a* = *kd* and *b* = *d* is replaced into the normalization formula to obtain the voltage *V*_2_ in the near-source area. The coordinate coefficient of the integral node is *k*_2*j*_ and the integral position parameter is *x*_2*j*_ = *k*_2*j*_·*d.* The integral formula in the near-source area is shown as below:(7)$$V_{2}=\int_{k.d}^{d}f\left(x\right)dx=\overset{n}{\underset{j=1}{\sum }}A_{2j}f\left(x_{2j}\right)$$
(8)$$x_{2j}=\frac{\left(1-k\right)d}{2}t_{2j}+\frac{\left(1+k\right)d}{2},j=1,2,3,\ldots \ldots ,\;n$$
(9)$$A_{2j}=\frac{\left(1-k\right)d}{n},j=1,2,3,\ldots \ldots ,n$$
where *V*_2_ is the numerical integration result of the interval *L*_2_; *n*, *A*_2*j*_, and *x*_2*j*_ refer to the amount, weights, and positions of the integral nodes in the interval *L*_2_ respectively; *k* is the reconstruction factor, and $k_{2j}=\;0.5\left[\left(1-k\right)t_{2j}+1+k\right]$ refers to the reconstruction coefficient in the *L*_2_.

Combining the reconstructed algorithm with the integral node interval partition, *m* and *n* integral nodes are taken in *L*_1_ and *L*_2_, respectively, to obtain the calculating model of the overhead line voltage in the whole integral path, which is shown as below:(10)$$V_{d}=V_{1}+V_{2}=\overset{m}{\underset{j=1}{\sum }}A_{1j}f\left(x_{1j}\right)+\overset{n}{\underset{j=1}{\sum }}A_{2j}f\left(x_{2j}\right)$$
where *V_d_* is the numerical integration result of the whole integral domain. 

According to the reconstructed algorithm model, the sensors with fewer nodes are installed in the near-ground area with less E-field distribution. Meanwhile, in the near-source area with intense E-field distribution that is near the overhead lines, there are relatively more sensors installed. Hence, the resource waste caused by inappropriate node position is avoided and the accuracy of voltage information restoration is guaranteed.

## 3. Simulation Calculation and Test

### 3.1. Judgment of Reconstruction Factors

The determination of the reconstruction factor *k* will directly determine the position of integral nodes, which is influenced by multiple factors, such as the voltage level and height of the overhead lines. The first-order differential variation rate of the E-field under the overhead lines and the distance *d* is proposed as the final criterion for segmentation, which can make the partition and linearization of the interval *L*_1_ and *L*_2_ more accurate.

The models of overhead lines of four voltage levels, including 10 kV, 35 kV, 66 kV, and 110 kV, are established in ANSYS Maxwell where the lead length is set as 2 m and lead radius is set as 6 mm. The height is set according to the standards, which is 6.5 m, 7 m, 7 m, and 7 m respectively with the increase of voltage levels. The ground is set as the reference surface of zero potential. Because all the overhead lines below 110 kV adopt the single lead, it is not necessary to consider the problem of splitting lead.

The model of wires show in Figure 4 were set in four levels of voltage and height, and the ground plane boundary parameter was set to five times the wire length, then, the three-phase time-varying voltage with a phase difference of 2 × π/3 was excited to the model. So, the E-field distribution information shown in Figure 5 was obtained.

It can be seen that the E-field information is concentrated in a small area under the wire. To make the E-field change trend more intuitive, its distribution data of the overhead line to the ground integration path is extracted. To further determine the integral node reconstruction factor, the E-field change rate parameter on the integral path is calculated by the 1st derivative of “Abs (Mag_E)”, and the result shown in Figure 6.

According to the minimum safety distance *d* for the leads, the heights are set in the interval of 6.5–7.0 m respectively. Then, the E-field values and the variation rate upon the partition sites can be calculated by the ANSYS Maxwell. According to the above criteria, the reconstruction factor *k* is distributed in the interval of [0.85, 0.90]. Through the simulation and calculation at specific voltage levels of overhead lines, the reconstruction factor *k* can be determined to acquire the interval partition standards of the node parameter reconstruction method.

As for the three-phase overhead line experiment platform adopted in the experiment, the reconstruction factor *k* equals to 0.875 through the aforementioned calculating method. The weights and node position of interval *L*_1_ and *L*_2_ after the node reconstruction through the Formulas (4)–(10) are shown as Table 2.

The E-field information of 400 data points in the integral path are compared and the interval lower limit values whose E-field variation rate exceeds 2 kV/m^2^ for the first time are used as the partition position. Then, the reconstruction factor and position parameter information for multi-level voltages of overhead lines are further obtained by simulations, and the results are shown in Table 3.

According to the results of Table 1, Table 2 and Table 3, after the reconstruction of the integral node parameters, the weights *A*_1*j*_/*A*_2*j*_ and positions *x*_1*j*_/*x*_2*j*_ of the integral nodes are related tightly to the height of the overhead lines, so the sensor configuration scheme is more suitable for the application to the multi-level voltages of overhead lines.

The node position in the near-ground region is significantly improved compared with that before the reconstruction. Since the E-field information changes slightly, one or two sensors such as *x*_11_ and *x*_21_ can be selected as the reference sensor to avoid waste of resources and reduce the impact comes from other interference. At the same time, reducing the distance between the near-ground and the near-source sensor satisfies the installation requirements that the sensor should keep a certain distance from the ground, which is conducive to the formation of the sensor array architecture.

Due to the abundance of E-field information, a multi-point sensor array can be set in the near-source region to achieve effective voltage reduction. The sensor array is more compact, denser, and convenient to install after the reconstruction, which can also improve the accuracy of the voltage measurement system. In the selection of the top node, it is necessary to select a reasonable near-source integration node configuration scheme based on the actual sensor insulation performance.

### 3.2. Simulation Test

In order to test the effect of the Gauss–Chebyshev algorithm after the node parameters reconstruction, a three-phase overhead line finite element E-field simulation model is established as shown in Figure 5. Among them, copper is used as the material for wires, and the wire radius is set to 7 mm and the length is 2 m. The distance between the wire and the ground is 1.5 m, while the phase spacing is 0.8 m, the ground potential boundary voltage is set to 0, the simulation duration is 20 ms, and the time step is 0.20 ms.

According to the judging method for the reconstruction factor *k* in Section 3.1, with 1.5 m height of the 10 kV wires, e.g., it can be judged that, in the range of 0–1.3125 m, the first-order differential of the E-field and the distance to the ground tends to be zero. The reconstructed node distribution is shown in Figure 7. Within the range of 1.3125–1.5 m, the E-field variation significantly increases. Thus, the reconstruction factor *k* is calculated about 0.875, which divides the integral intervals into the *L*_1_ and *L*_2_.

As for the whole integral interval, taking the sensor installation error and the possible E-field distortion into account, in the case of guarantee the integral measuring accuracy, the quantity of sensors installed should be reduced as far as possible. Therefore, in the simulation model, the node distribution scheme where *m* = 1 and *n* = 2 as shown in Table 4 is adopted.

Based on the E-field simulation results according to Section 3.1, the E-field change information of the three nodes with simulation time of 20 ms is extracted as shown in Figure 8.

The field calculator is used to finish the calculation of Formula (10), through which the integral results based on weighted sum of E-field parameters under the distribution scheme where *m* = 1 and *n* = 2 are obtained.

It can be observed from Figure 9 that the amplitude and frequency error of the simulation measurement result has a small partitioned node configuration scheme. The integral calculating results at the peak moment of 5 ms and 15 ms are taken to conduct the measuring deviation evaluation on the algorithm and the obtained measuring error is about 0.37%, which proves that the node improvement method owns the relatively high measuring accuracy from the perspective of simulation.

## 4. Experiment and Discussion

The simulation experiment results show that the improved voltage measurement system based on the Gauss–Chebyshev algorithm has good measurement effect. To verify the effectiveness of the algorithm, according to the aforementioned voltage measurement system structure, a test platform for three-phase horizontally arranged overhead lines with adjustable voltage in the range of 0–30 kV is built to test the accuracy of the voltage measuring system. The measurement platform is shown in Figure 10. In order to capture the maximum E-field information, the axial lead of the E-field sensor array is located on the integral path directly below the A-phase wire, and the processed signal is transmitted to LabVIEW software by the wireless communication module to calculate and display the integral result of the algorithm.

The Plexiglas strut with the standard millimeter scale and insulated long-arm support are adopted to reduce the errors caused by the sensor installation. The high-voltage probe is placed at the input end of the A-phase overhead line, and the sensor array is placed at the other end to reduce the interference of E-field caused by the other media. Hence, the position error caused by the sensor installation can be reduced by the precise fixing of sensors.

After applying a 20-kV voltage excitation through the ramp-up platform, the sensor output waveform at the 3-node positions are detected and compared with the standard high-voltage probe waveform to verify the quality of each output waveform as shown in Figure 11.

In the LabVIEW software, the algorithm routine of Gauss–Chebyshev voltage measurement system is designed by adjusting the integral weight and the node position values. The parameters such as sampling rate, local port, and IP address are set correctly. The E-field information collected by the sensor of each node is weighted and summed under the node configuration schemes of the Gauss–Chebyshev algorithm and the integral node parameters reconstruction method respectively to compare the measuring accuracy of these two methods.

The Figure 12 shows the final measurement result of the node parameters reconstruction method at 10 kV. The yellow, blue, and green waveforms respectively correspond to the E-field information detected by different nodes of sensor, and the white waveform refers to numerical integration results obtained by weighted summation of E-field parameters at the position of three nodes.

The voltage of the overhead line is further adjusted through the booster, and a plurality of tests is carried out in the voltage range of 10–20 kV. The measured voltage peak-to-peak value of the high-voltage probe is recorded as *V*_r_. In the LabVIEW Integral Program, *V*_c_ represents the integral computing voltage calculated by the Gauss–Chebyshev algorithm, and *V*_n_ represents that value calculated by the node parameters reconstruction method of the Gauss–Chebyshev algorithm. The following experimental results shown in Table 5 are obtained by comparing the measured data of the two methods:

The following conclusions can be drawn by combining the algorithm model, simulation, and experiment comparison:(1)The parallel distributed D-dot E-field sensor has good output characteristics, and the phase deviation of the E-field signal of each node is less than 1.8°, which ensures the reliability of the E-field parameters at the position of the reconstructed node.(2)In the node parameter reconstruction simulation calculation model, the weights and the position parameters of the integral nodes are determined by the voltage and height of overhead lines, so the integral algorithm model can match with the overhead line parameters well.(3)Similar to the simulation test results, the deviation of the voltage measurement system after reconstruction is less than 0.4% in the voltage range of 10–20 kV, which is nearly 10 times smaller than the deviation of the system before the node reconstruction.(4)The E-field waveform, the simulated calculation waveform, and the platform measurement waveform have good consistency, which proves that the reconstructed measurement system is featured by good anti-interference performance.

Comparing with the previous researches, the node parameters reconstruction method of the Gauss–Chebyshev algorithm can match the integral node parameters with the overhead lines, optimize distribution schemes of sensors, and improve the stability of the contactless voltage measurement system on the premise of ensuring the good measurement accuracy. Therefore, the node reconstructed Gauss–Chebyshev algorithm is more applicable for promotion.

## 5. Conclusions

Aiming at the problems existing in the contactless E-field integral voltage measurement method, such as the improper distribution of nodes and unstable measurement accuracy, a new type of parallel distributed electrode D-dot sensor was adapted to the construction of a voltage measurement system. Then, the research focused on the reconstruction model of the integral node algorithm and the comprehensive determination method for the reconstruction factor. Through simulations and system tests, the waveform reduction effect and system measurement accuracy before and after parameter reconstruction of the Gauss–Chebyshev integral node were verified. The results illustrate that the D-dot contactless voltage measurement system through the parameter reconstruction of integral nodes can reduce the distortion obviously and restore the overhead line voltage information effectively. This paper solves the problem with parameter matching between the integral nodes and the overhead lines which exists in the E-field integral method, reduces the external interference through the node parameter reconstruction, and makes the sensor distribution scheme more conducive to practical application and improves the voltage measurement stability of the system. Next, we will continue to explore the transient signal monitoring terminal and its measurement algorithms to predict the potential malfunction of overhead lines.

## Figures and Tables

**Figure 1 sensors-20-00246-f001:**
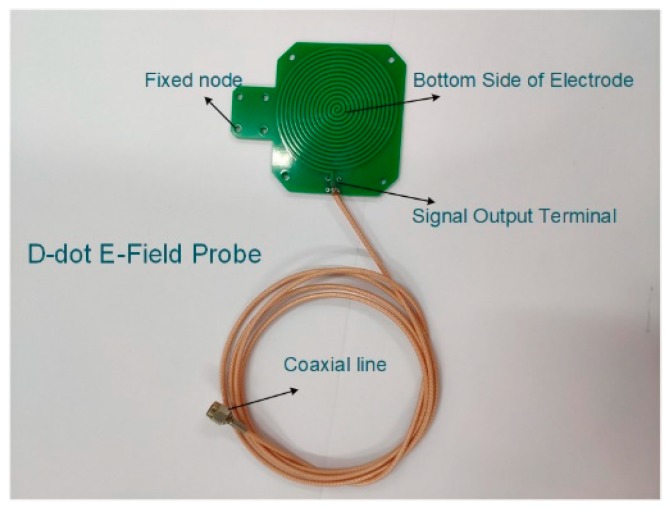
D-dot E-field sensor with parallel distributed electrode.

**Figure 2 sensors-20-00246-f002:**
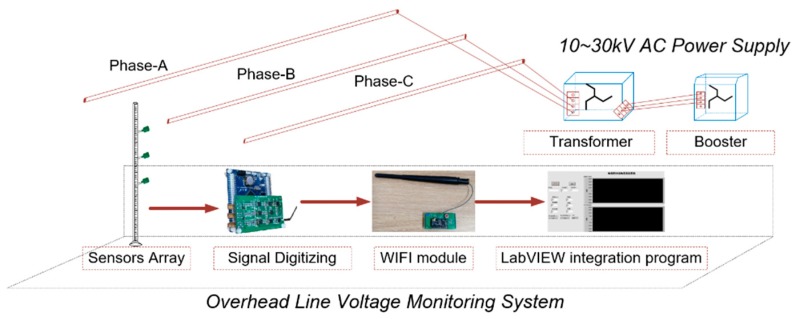
Framework of the overhead line voltage measurement system.

**Figure 3 sensors-20-00246-f003:**
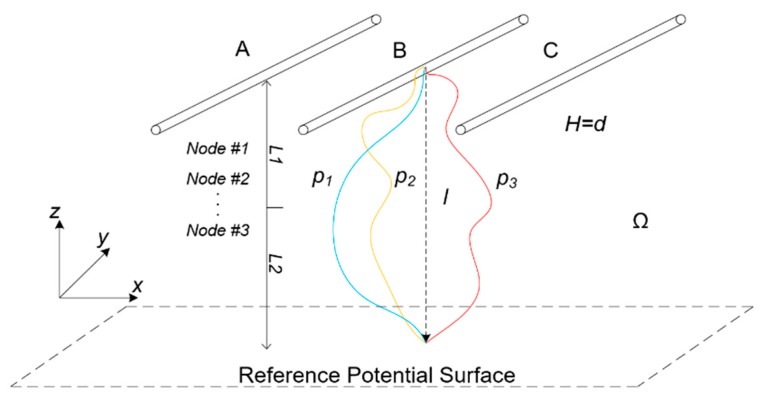
Schematic of the Gauss–Chebyshev integral calculation.

**Figure 4 sensors-20-00246-f004:**
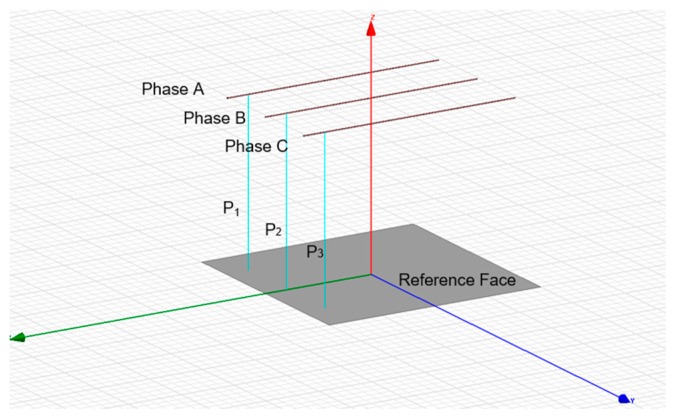
Three-phase overhead line simulation model.

**Figure 5 sensors-20-00246-f005:**
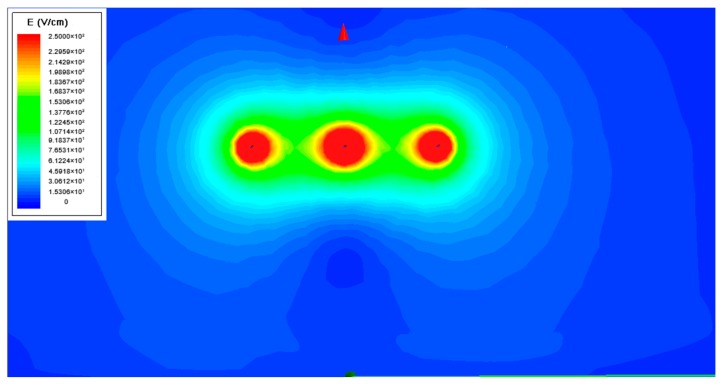
Distribution of E-field below the 10 kV three-phase overhead line.

**Figure 6 sensors-20-00246-f006:**
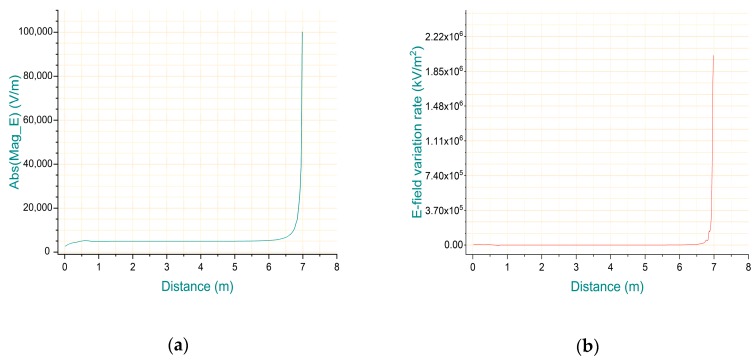
E-field information under 35 kV overhead line: (**a**) the distribution of E-field; (**b**) the E-field variation rate tendency.

**Figure 7 sensors-20-00246-f007:**
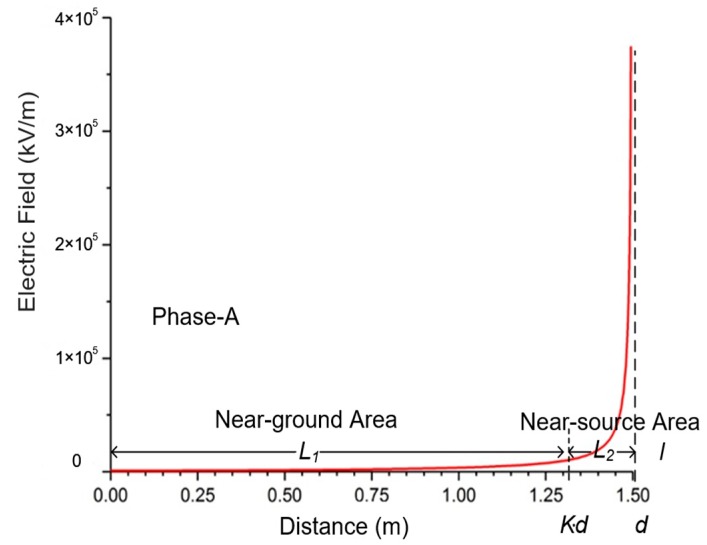
Simulation integral interval partition result.

**Figure 8 sensors-20-00246-f008:**
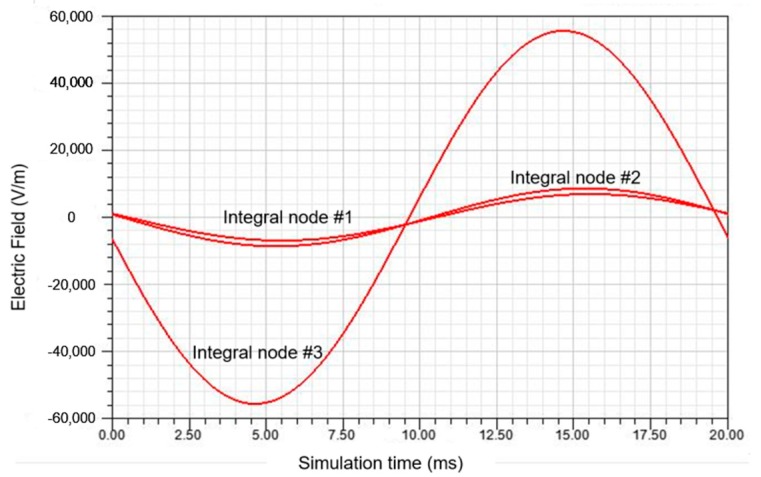
E-field waveform at reconstructed node position.

**Figure 9 sensors-20-00246-f009:**
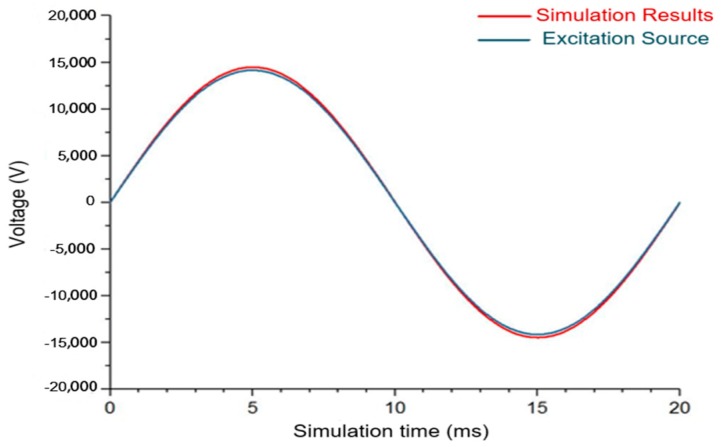
Calculating results of the node reconstruction method.

**Figure 10 sensors-20-00246-f010:**
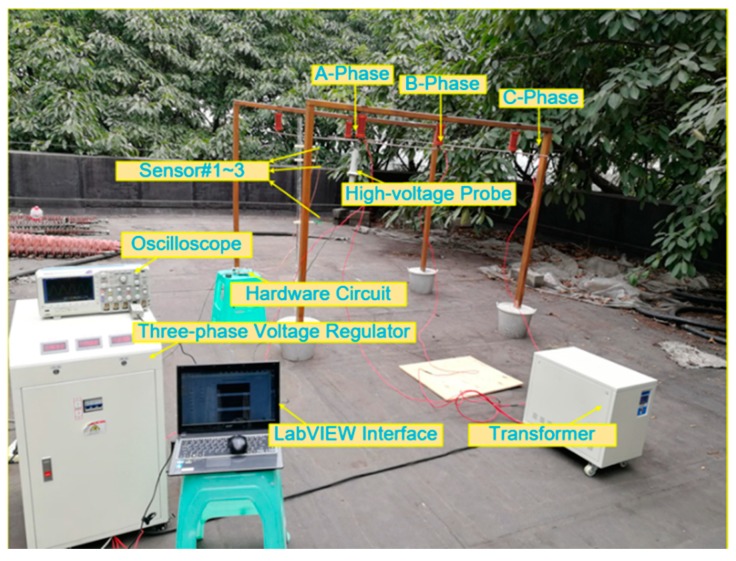
Three-phase overhead line voltage measurement test platform.

**Figure 11 sensors-20-00246-f011:**
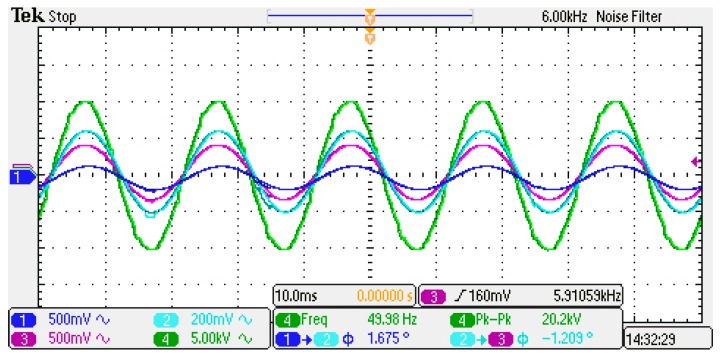
Waveform measured by sensors under the 20-kV voltage excitation.

**Figure 12 sensors-20-00246-f012:**
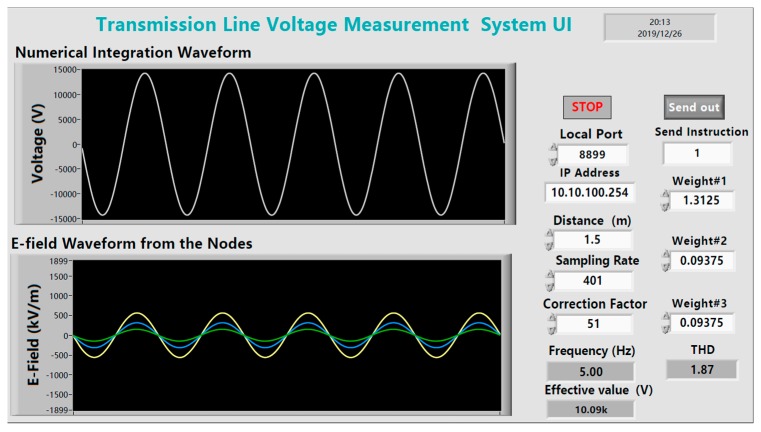
Measurement results displayed in the LabVIEW UI.

**Table 1 sensors-20-00246-t001:** Integral node information of Gauss–Chebyshev algorithm.

*n*	*A_n_*	*t*_1_/*x*_1_	*t*_2_/*x*_2_	*t*_3_/*x*_3_	*t*_4_/*x*_4_	*t*_5_/*x*_5_
2	1.5708	0.7071/0.8535*d*	−0.7071/0.1465*d*			
3	1.0472	0.8660/0.9330*d*	0/0.5000*d*	−0.8660/0.0670*d*		
4	0.785	0.9239/0.9620*d*	0.3827/0.6914*d*	−0.3827/0.3087*d*	−0.9239/0.0381*d*	
5	0.628	0.9755/0.9878*d*	0.5878/0.7939*d*	0/0.5000*d*	−0.5878/0.2061*d*	−0.9755/0.0245*d*

**Table 2 sensors-20-00246-t002:** The coefficient of integral nodes in the near-ground *L*_1_ and the near-source *L*_2_.

*m*	*A*_1*j*_/*A*_2*j*_	*x*_11_/*x*_21_	*x*_12_/*x*_22_	*x*_13_/*x*_23_	*x*_14_/*x*_24_
1	0.8750*d*/0.1250*d*	0.4375*d*/0.9375*d*			
2	0.4375*d*/0.0625*d*	0.1849*d*/0.9014*d*	0.6901*d*/0.9736*d*		
3	0.2917*d*/0.0417*d*	0.1281*d*/0.8933*d*	0.4375*d*/0.9375*d*	0.7469*d*/0.9817*d*	
4	0.2188*d*/0.0313*d*	0.0898*d*/0.8878*d*	0.3554*d*/0.9258*d*	0.5196*d*/0.9492*d*	0.7852*d*/0.9872*d*

**Table 3 sensors-20-00246-t003:** The reconstruction factor and position parameters of overhead lines.

Voltage Level *V_L_* (kV)	Lead Height *d* (m)	Partition Position (m)	E-Field Values upon the Partition Sites (V/m)	E-Field Variation Rate upon the Partition Sites (kV/m^2^)	Reconstruction Factor *k*
10	6.5	5.785	1672.044	2.131	0.890
35	7.0	6.175	5481.911	2.358	0.884
66	7.0	6.035	9955.669	2.753	0.862
110	7.0	5.982	16,429.093	2.224	0.854

**Table 4 sensors-20-00246-t004:** Node reconstruction configuration scheme.

Interval	Node No.	Position (m)	Weights
*L* _1_	1	0.656	1.3125
*L* _2_	2	1.352	0.09375
	3	1.460	0.09375

**Table 5 sensors-20-00246-t005:** Gauss–Chebyshev node parameter reconstruction method measurement results.

*V*_r_ (kV)	*V*_c_ (kV)	ε%	*V*_n_ (kV)	ε%
10.1	9.74	3.564	10.06	0.396
11.9	11.55	2.941	11.86	0.337
14.1	13.49	4.326	14.05	0.355
16.0	15.45	3.438	15.94	0.375
18.1	17.73	2.044	18.03	0.387
20.2	19.6	2.970	20.13	0.347

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
