# Peer review of "Simulation and Test of a Contactless Voltage Measurement Method for Overhead Lines Based on Reconstruction of Integral Node Parameters"

_sensors, 2019, doi:10.3390/s20010246_

Round 1

Reviewer 1 Report

Contactless voltage measurement is a import techniques for the online monitoring and condition diagnosis of overhead line. The key is to improve its accuracy for onsite application. The authors have studied several algorithm to restore the transmission line voltage, such as Gauss-Legendre, Guass Integral and Chebyshev algorithm, and published articles about this topic in 2018, 2019 [ref. 27,28,29,32]. Here,I have three questions as below; 1) From Ref. 29 (Sensor, 2018), the relative error is less than 0.27% (10kV~20kV) by Gaussian Integral method. In this paper, the relative error is more than 0.347% by Gauss-Chebyshev algorithm. The latter is worse. Then, what is the contribution of this paper?

(2) In Line 333-335, the third conclusion: "..., which is about 10 times higher than the system accuracy measured before parameters reconstruction" . What does the phrase "system accuracy measured before parameters reconstruction" mean? Please give the accurate data of the system accuracy.

(3) Since the authors have evaluated more than 3 similar algorithm upon the same voltage measurement system and have accumulated a wealth of experience. Could the authors give a clear expression to let the readers know that which method is the best?

I suggest that the numeric comparison of measurement errors based on different reconstruction methods could be provided in the end of section 4.

Reviewer 2 Report

The article is interesting and after the corrections is suitable for publication in Sensors.

Round 2

Reviewer 1 Report

I think the authors have answered my questions.

Author Response

Dear reviewer,

Thank you again for your affirmation of our research. Your previous suggestions have greatly helped us, and we will continue to improve our research. Wish you all the best!

Yours sincerely

Xiaojun Yan

Reviewer 2 Report

The article is interesting and after the corrections is suitable for publication in Sensors.The reviewer has some minor comments contained in the detached file.
